

# A lightweight convolutional neural network (CNN) model for diatom classification: DiatomNet

Huseyin Gunduz and Serkan Gunal

Department of Computer Engineering, Eskisehir Technical University, Eskisehir, Turkiye

## ABSTRACT

Diatoms are a type of algae with many species. Accurate and quick classification of diatom species is important in many fields, such as water quality analysis and weather change forecasting. Traditional methods for diatom classification, specifically morphological taxonomy and molecular detection, are time-consuming and may not provide satisfactory performance. However, in recent years, deep learning has demonstrated impressive performance in this task, just like other image classification problems. On the other hand, networks with more layers do not guarantee increased accuracy. While increasing depth can be useful in capturing complex features and patterns, it also introduces challenges such as vanishing gradients, overfitting, and optimization challenges. Therefore, in our work, we propose DiatomNet, a lightweight convolutional neural network (CNN) model that can classify diatom species accurately while requiring low computing resources. A recently introduced dataset consisting of 3,027 diatom images and 68 diatom species is used to train and evaluate the model. The model is compared with well-known and successful CNN models (*i.e.*, AlexNet, GoogleNet, Inceptionv3, ResNet18, VGG16, and Xception) and their customized versions obtained with transfer learning. The comparison is based on several success metrics: accuracy, precision, recall, F-measure, number of learnable parameters, training, and prediction time. Eventually, the experimental results reveal that DiatomNet outperforms the other models regarding all metrics with just a few exceptions. Therefore, it is a lightweight but strong candidate for diatom classification tasks.

# INTRODUCTION

Diatoms are one of the main groups of eukaryotic algae and one of the leading phytoplankton. The most important feature of these small microorganisms is that they have two covers made of asymmetric and silica. These covers are very robust and have a porous structure. Therefore, they are very durable compared to other living skeletons.

There are 250 genera and approximately 100,000 species of diatoms distributed worldwide (*Round, Crawford & Mann, 1990*). Diatoms can be found in seas, freshwater, soil, and humid areas. The vast majority of diatoms live deep in water. However, some of them can live on the surface of the water. It is essential for oceans that diatom populations

Corresponding author
Huseyin Gunduz,
huseyingunduz@anadolu.edu.tr

are much larger than other living groups because these creatures work as the primary producers of the oceans.

In recent years, the number of research on diatoms has increased by revealing the relationships between diatoms and environmental factors (*Truchy et al., 2022*; *Antonija et al., 2023*). In particular, the changes in diatom diversity are used for different research purposes, such as climate change and water quality assessment. To this end, the classification of diatoms has become very important. Also, the amount and variety of diatoms in a region can be used to estimate the environmental background of water sources in that region (*Archibald, 1972*; *Kong, 2022*).

Diatoms are the only organisms with cell walls composed of transparent, opaline silica. The cell walls of diatoms are decorated with intricate and striking patterns of silica. Despite their geometrically similar appearances, most diatom species are distinguished by these patterns. Due to the wide variety of diatoms, manual classification of them is a challenging as well as a time-consuming task. For this reason, automated approaches mostly based on image processing and machine learning methods have emerged. The first annotated dataset for the automatic classification of diatoms was proposed in *Du Buf et al. (1999)*. In that study, classification was done using an unsupervised model, Automatic Diatom Identification and Classification (ADIAC). For classifying diatoms, the most appropriate diatom resemblance was determined using the graphical valve definitions of the diatoms. In *Bayer & Du Buf (2002)*, geometrical, textural, morphological, and frequency descriptors were obtained from diatom images and classified with decision trees. In *Luo et al. (2010)*, only round diatoms were considered, and the classification was carried out with the help of artificial neural networks using texture features containing the spectrum information. In *Dimitrovski et al. (2012)*, a hierarchical multi-label classification model was proposed to classify diatoms. In *Kloster, Kauer & Beszteri (2014)*, diatoms were classified by extracting the shape attributes with elliptic Fourier transform and comparing those attributes to those of the diatom templates. In *Bueno et al. (2017)*, local binary pattern (*Ahonen, Hadid & Pietikäinen, 2006*) and log-Gabor (*Fischer et al., 2007*) features, as well as some morphological and statistical features, were used with a bagging decision tree to classify diatoms. In *Sánchez, Cristóbal & Bueno (2019)*, proposed to characterize the diatom life cycle by the contour shape and the texture features that change during the algae life cycle Gabor filters were used to describe the inner ornamentation and Elliptical Fourier Descriptors were used to describe the diatom contour while phasing congruency. Then, they used several supervised and unsupervised learning techniques to classify diatoms.

In recent years, for the classification of diatoms, deep learning-based methods have emerged, as well. In *Pedraza et al. (2017)*, deep neural networks were used for diatom classification rather than traditional feature engineering approaches. *Libreros et al. (2019)* detected diatoms based on the combination of Scale and Curvature Invariant Ridge Detector followed by a post-processing method and the nested convolutional neural network (CNN) to classify diatom genus. *Kloster et al. (2020)* proposed a CNN-based method for taxonomic classification of microalgal groups of diatoms. They used microscopic slides of diatoms and applied some techniques to obtain a high-quality image, then a focus stacking technique to obtain the focus-enhanced image. Slide stitching

combines these highly enhanced image regions into gigapixel-sized virtual slides. Then, they annotated the virtual slides and classified them with CNNs. *Chaushevska et al. (2020)* applied transfer learning using a pre-trained Inceptionv3 model and extracted features to train a tree-ensemble classifier. They used ensembles of predictive clustering trees for a hierarchical multi-label diatom classification. In *Pu et al. (2023)*, to identify diatoms, collected diatom images and employed deep learning techniques, utilizing the ResNet50, ResNet152, MobileNetV2, and VGG16 networks. Additionally, their method, which incorporates model prediction and cosine similarity, enhances accuracy in low-probability predictions.

Though existing CNN models provide satisfactory performance in classifying diatom images, their computational load is significantly high due to the complexity of network architectures. Therefore, in our work, DiatomNet, a lightweight but strong CNN model, is proposed for automated diatom image classification. The depth and number of learnable parameters of DiatomNet are significantly small compared to well-known CNN models such as AlexNet, GoogleNet, Inceptionv3, ResNet18, VGG16, and Xception. To validate the efficiency of DiatomNet, an extensive set of experimental work was conducted using a recently introduced dataset consisting of 3,027 diatom images and 68 diatom species. Several success metrics were utilized during the experiments, including accuracy, precision, recall, F-measure, number of learnable parameters, training time, and prediction time. The results of the experimental work verify that DiatomNet, in most cases, outperforms the other CNN models and their customized versions obtained with transfer learning on the pre-trained versions of these models.

The rest of this article is organized as follows. In "Material and Methods", the diatom image dataset is described, convolutional neural networks and transfer learning are briefly explained, and DiatomNet is introduced. In "Experimental Work", the experimental work is described, the results, discussions and, a comparison with the literature are given. Finally, some concluding remarks and future work are given in "Conclusions".

## MATERIALS AND METHODS

In this section, the diatom image dataset, which is used to train and evaluate the proposed model, is first described. Then, the architecture of typical CNNs and the transfer learning approach are briefly explained. Finally, DiatomNet is introduced, the proposed lightweight CNN architecture for diatom classification.

### Dataset

A recently introduced diatom image dataset is used to evaluate the proposed model. The dataset consists of 3,027 images of a total of 68 diatom species commonly found in the rivers of Turkiye (*Gunduz, Solak & Gunal, 2022*). The spatial resolution of the images in the dataset is 2,112 × 1,584 pixels. The boundaries of all diatoms in each image are annotated at the pixel level and several diatom experts identify the species of the diatoms. Sample diatom images from the dataset as well as their annotations are shown in Fig. 1. The distribution of the diatom species in the dataset is summarized in Table 1.

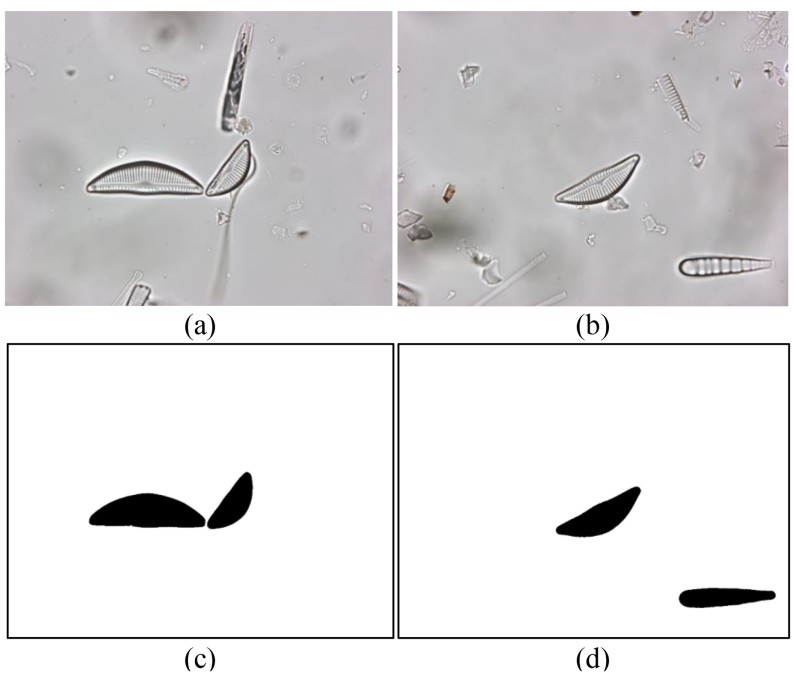

**Figure 1** (A and B) Sample diatom images from the dataset and (C and D) their annotations.

**Table 1 Distribution of the diatom species in the dataset.**

| No. | Diatom species | #Diatoms | No. | Diatom species | #Diatoms |
|---|---|---|---|---|---|
| 1 | *Achnanthidium biasolettianum* | 60 | 35 | *Halamphora veneta* | 40 |
| 2 | *Achnanthidium minutissimum* | 13 | 36 | *Hantzschiana abundans* | 10 |
| 3 | *Adlafia minuscula* | 26 | 37 | *Humidophila contenta* | 53 |
| 4 | *Amphora inariensis* | 6 | 38 | *Humidophila perpusilla* | 35 |
| 5 | *Amphora pediculus* | 22 | 39 | *Luticola nivalis* | 7 |
| 6 | *Caloneis lancettula* | 10 | 40 | *Meridion circulare* | 65 |
| 7 | *Cocconeis pseudolineata* | 49 | 41 | *Navicula capitatoradiata* | 15 |
| 8 | *Cymbella cantonatii* | 54 | 42 | *Navicula cryptocephala* | 24 |
| 9 | *Cymbella excisa* | 59 | 43 | *Navicula cryptotenella* | 272 |
| 10 | *Cymbella excisa var. procera* | 6 | 44 | *Navicula cryptotenelloides* | 77 |
| 11 | *Cymbella excisa var. subcapitata* | 44 | 45 | *Navicula gregaria* | 22 |
| 12 | *Cymbopleura amphicephala* | 6 | 46 | *Navicula lanceolata* | 9 |
| 13 | *Denticula kuetzingii* | 14 | 47 | *Navicula moskalii* | 19 |
| 14 | *Diatoma mesodon* | 53 | 48 | *Navicula novaesiberica* | 9 |
| 15 | *Diatoma moniliformis* | 56 | 49 | *Navicula reichardtiana* | 165 |
| 16 | *Didymosphenia geminata* | 6 | 50 | *Navicula tripunctata* | 22 |
| 17 | *Diploneis fontanella* | 8 | 51 | *Navicula trivialis* | 34 |
| 18 | *Encyonema silesiacum* | 182 | 52 | *Navicula upsaliensis* | 45 |
| 19 | *Encyonema ventricosum* | 61 | 53 | *Neidiomorpha binodiformis* | 6 |
| 20 | *Epithemia argus* | 17 | 54 | *Nitzschia archibaldii* | 19 |

| No. | Diatom species | #Diatoms | No. | Diatom species | #Diatoms |
|-----|----------------|----------|-----|----------------|----------|
| 21 | *Epithemia goeppertiana* | 7 | 55 | *Nitzschia hantzschiana* | 30 |
| 22 | *Fragilaria recapitellata* | 184 | 56 | *Nitzschia linearis* | 36 |
| 23 | *Frustulia vulgaris* | 12 | 57 | *Nitzschia palea* | 6 |
| 24 | *Gomphonema calcifugum* | 39 | 58 | *Nitzschia recta* | 7 |
| 25 | *Gomphonema drutelingense* | 15 | 59 | *Pantocsekiella ocellata* | 27 |
| 26 | *Gomphonema exilissimum* | 8 | 60 | *Pinnularia brebissonii* | 36 |
| 27 | *Gomphonema micropus* | 16 | 61 | *Planothidium frequentissimum* | 26 |
| 28 | *Gomphonema minutum* | 15 | 62 | *Planothidium lanceolatum* | 132 |
| 29 | *Gomphonema olivaceum* | 386 | 63 | *Rhoicosphenia abbreviata* | 77 |
| 30 | *Gomphonema pumilum* | 11 | 64 | *Sellaphora radiosa* | 9 |
| 31 | *Gomphonema pumilum var. rigidum* | 33 | 65 | *Sellaphora saugerresii* | 6 |
| 32 | *Gomphonema supertergestinum* | 14 | 66 | *Stauroneis blazenciciae* | 7 |
| 33 | *Gomphonema tergestinum* | 85 | 67 | *Surella minuta* | 6 |
| 34 | *Halamphora paraveneta* | 33 | 68 | *Surirella brebissonii var. kuetzingii* | 64 |

All diatoms in the dataset are extracted using their annotations and then normalized horizontally. Sample diatom images obtained after the preprocessing step are shown in Fig. 2. The number of images for each diatom class in the dataset is listed in Table 1.

The dataset is further expanded by applying several augmentation techniques, including vertical flip, horizontal flip, and vertical-horizontal flip. In this way, the augmented dataset, which is four times larger than the original one, is obtained. Since the sizes of the images in the dataset vary, each image is also resized to 432 × 128 pixels, corresponding to the average width by the average height of all images in the dataset. An example of the resized diatom images and their augmented versions are shown in Fig. 3.

The original and augmented datasets are split into three parts: 70% training, 15% validation, and 15% test per class. The images are randomly selected with stratified sampling to ensure the representation of every class in each split. The distributions of the images on each part of the datasets are listed in Table 2.

## Convolutional neural networks

A CNN is basically the regularized version of a multilayer perceptron. A typical CNN mainly consists of an input layer, several convolutional and pooling layers, and a fully connected layer. The input layer receives the input. The convolutional and pooling layers perform the feature learning task, and the classification is realized in the fully connected layer. In CNNs, various pooling methods, ReLU activation functions, and soft-max activation functions can be used to obtain different architectures. An optimization algorithm in the training phase learns unknown parameters of a CNN. The performance of a CNN depends on the size and balance of the training dataset as well as the computational resources on which the CNN is trained.

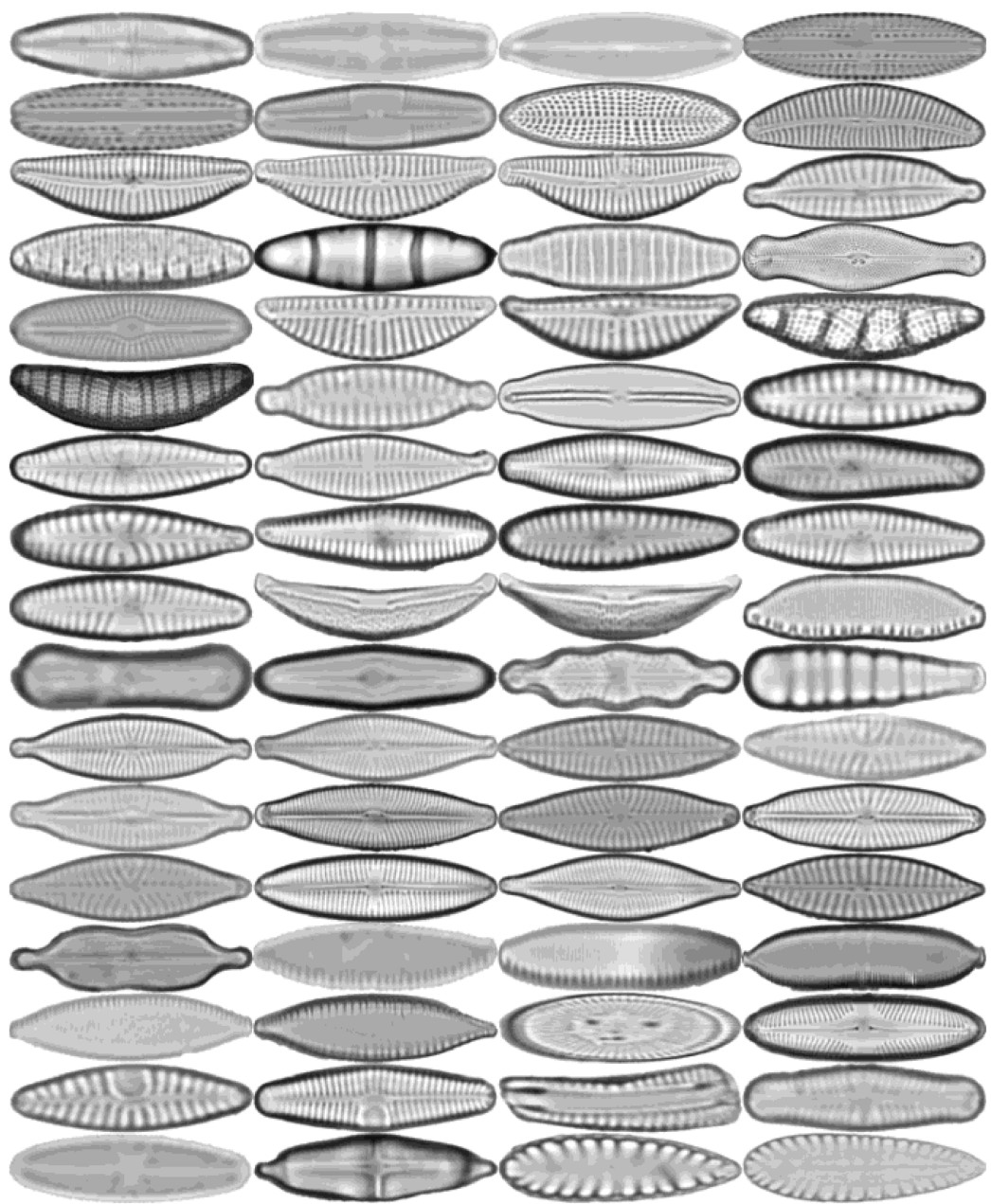

**Figure 2 Sample images for each diatom species.** The images are ordered from left to right and top to bottom according to their class numbers from 1 to 68.

CNNs offer several advantages over other classification methods. One major advantage of CNNs is that they automatically extract features from images without needing pre-extraction. These features are then optimally adjusted to produce the desired result. Additionally, CNNs are robust to noise and can recognize patterns in distorted images. Transfer learning is also supported by CNNs, which means they can be trained on one task and used for another with little or no additional training. Another key benefit of CNNs is that they can learn to recognize complex image patterns by analyzing large datasets to achieve significantly high accuracy rates.

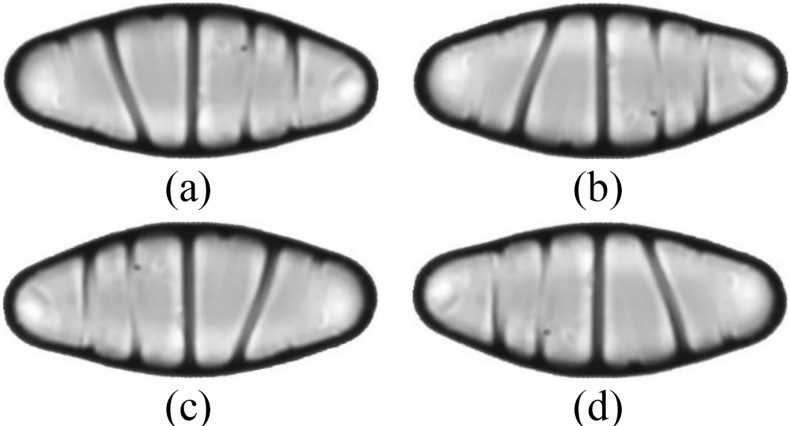

**Figure 3** (A) A resized diatom image and its augmented versions with (B) vertical flip, (C) horizontal flip, and (D) vertical-horizontal flip.

**Table 2 The number of images in the original and augmented datasets.**

|  | Original | Augmented |
| --- | --- | --- |
| Train | 2,115 | 8,479 |
| Validation | 459 | 1,814 |
| Test | 453 | 1,815 |

However, it's important to note that CNNs also have some limitations. Due to their complex architecture and numerous layers and parameters require a significant amount of memory and processing power to train and run. This makes them computationally expensive, which may not be feasible for all applications. Additionally, to achieve high accuracy rates, CNNs require large datasets with many examples of the patterns they are being trained to recognize. If the dataset is too small, the CNN may overfit the training data and perform poorly on new, unseen data.

In this work, in addition to the proposed CNN model, six well-known CNN models, including AlexNet (*Krizhevsky, Sutskever & Hinton, 2012*), VGG16 (*Simonyan & Zisserman, 2015*), GoogleNet (*Szegedy et al., 2015*), ResNet18 (*He et al., 2016*), Inception (*Szegedy et al., 2016*), and Xception (*Chollet, 2016*) are also used for comparison purposes. AlexNet consists of eight layers. Five of them are convolutional layers and the others are fully connected. AlexNet has been trained with 1,000 class labels available in the ImageNet dataset (*Krizhevsky, Sutskever & Hinton, 2012*). The main feature of a VGG network is that the depth of the network is increased using a small filter size of 3 × 3 (*Simonyan & Zisserman, 2015*). VGG16 consists of 16 convolutional layers. The input image size of VGG is 224 × 224 and the fully connected layer is designed to classify 1,000 classes as described in *Simonyan & Zisserman (2015)*. GoogleNet is the architecture used in the ILSVRC 2014 competition. This network consists of 22 layers and uses average pooling before the classification. GoogleNet also uses a linear layer before the softmax layer to easily adapt the network to other label sets (*Szegedy et al., 2015*). ResNet is the winner of the 2015 ILSVRC.

This network presents a residual learning framework, and the layers are reformulated as residual functions. The ResNet18 architecture contains 18 layers. Inceptionv3 has 48 layers, uses label smoothing, and factorizes 7 × 7 convolutions and an auxiliary classifier (*Szegedy et al., 2016*). Xception architecture, inspired by Inception, uses depth-wise separable convolutions instead of inception modules (*Chollet, 2016*). This model might offer better performance than Inceptionv3, especially with large datasets.

## Transfer learning

As the previously explained CNN models can be trained from the scratch for a given image classification task, their pre-trained versions on a significantly large image dataset, for example ImageNet (*Deng et al., 2009*), can be directly used, as well. However, pre-trained models may not work well for certain classes of images that are relatively rare in the training dataset. In such cases, a pre-trained model can be customized with the help of transfer learning to be able to classify images belonging to new classes, as previously described in *Pan & Yang (2010)*, *Kaplan Berkaya, Sora Gunal & Gunal (2021)*, *Messaoudi et al. (2023)*. Thus, well-learned feature representations of a pre-trained network can be utilized to classify new images with a small amount of training data. To perform this process, classification and fully-connected layers at the end of the pre-trained network are substituted, aligning with the number of classes for the new classification task. Subsequently, the adapted pre-trained network undergoes re-training with the dataset pertaining to the recently introduced classes. Consequently, as opposed to the construction of a new network from scratch, this method improves both classification performance and training time, demanding a notably small training dataset. Due to these advantages of the transfer learning approach, in this work, the performances of the pre-trained CNN models with transfer learning are also analyzed and compared with our proposed model.

## Proposed model: DiatomNet

In recent years, the successful results of deep learning in image classification have led researchers to strive for better accuracy. This therefore gave rise to deep architectures with millions of parameters (*Krizhevsky, Sutskever & Hinton, 2012*; *Simonyan & Zisserman, 2015*). However, simply increasing depth does not always lead to improved accuracy (*He & Jian, 2015*).

The success of networks may not increase with deeper architectures. Although deeper networks can capture complex features, they also introduce challenges such as vanishing gradients, overfitting, and optimization challenges.

As networks become deeper, gradients in backpropagation may become very small, and in this case, learning may slow down or stop. This is the vanishing gradient problem, which hinders the training of deep networks. Deeper networks with more parameters are susceptible to overfitting, especially with smaller training datasets. While overfitting allows the model to learn from training data, including noise and outliers, it causes it to have difficulty generalizing to unseen data.

Deeper networks are harder to optimize. Effective optimization techniques and appropriate weight initialization are crucial to ensure that the model converges to a good

**Table 3 Parameters of the DiatomNet architecture.**

| Type | Patch size/stride | Output size | Depth | #1 × 1 | #3 × 3 reduce | #3 × 3 | #5 × 5 reduce | #5 × 5 | Pool Proj |
|---|---|---|---|---|---|---|---|---|---|
| Convolutional | 7 × 7/2 | 216 × 64 × 64 | 1 | – | – | – | – | – | – |
| Max pool | 3 × 3/2 | 108 × 32 × 64 | 0 | – | – | – | – | – | – |
| Convolutional | 3 × 3/1 | 108 × 32 × 192 | 2 | – | 64 | – | – | – | – |
| Max pool | 3 × 3/2 | 54 × 16 × 192 | 0 | – | – | – | – | – | – |
| Inception | – | 54 × 16 × 256 | 2 | 64 | 96 | 128 | 16 | 32 | 32 |
| Max pool | 3 × 3/2 | 27 × 8 × 256 | 0 | – | – | – | – | – | – |
| Inception | – | 27 × 8 × 512 | 2 | 192 | 96 | 208 | 16 | 48 | 64 |
| Max pool | 3 × 3/2 | 13 × 4 × 512 | 0 | – | – | – | – | – | – |
| Inception | – | 6 × 2 × 1,024 | 2 | 384 | 192 | 384 | 48 | 128 | 128 |
| Avg pool | 6 × 2/1 | 1 × 1 × 1,024 | 0 | – | – | – | – | – | – |
| Dropout (40%) | – | 1 × 1 × 1,024 | 0 | – | – | – | – | – | – |
| Linear | – | 1 × 1 × 68 | 1 | – | – | – | – | – | – |
| Softmax | – | 1 × 1 × 68 | 0 | – | – | – | – | – | – |

solution. Deeper networks require more computational resources for both training and inference. It can be a limitation in situations where limited computing power is available, such as embedded systems or mobile devices.

Beyond a certain depth, accuracy may not increase as computational costs continue to increase. This concept, known as the law of diminishing returns, suggests that adding more layers provides diminishing benefits in terms of performance gains.

In this study, we propose DiatomNet, a new lightweight CNN architecture for the diatom classification task, to overcome the above-mentioned problems. The layers and parameters of DiatomNet are depicted in Table 3 and Fig. 4. DiatomNet is a variant of the Inception network (Szegedy et al., 2015). The area occupied by the object in each image may be different. Therefore, the choice of filter size is very important in the convolutional layer. With a large filter size, global information can be gathered. On the other hand, a smaller filter size can get the local information. Therefore, in the Inception networks, the convolution is performed using different filter sizes in each inception module. In the literature, there are different versions of the Inception network (Szegedy et al., 2015, 2016, 2017). While a popular inception-based network, GoogleNet, has nine inception modules, DiatomNet has only three inception modules. While designing DiatomNet, it was experimentally verified that only three inception modules would be enough and provide satisfactory performance with less complexity. Before the inception modules, 7 × 7 convolutions and 3 × 3 max-pooling are used. After cross-channel normalization, 3 × 3 convolutions and 3 × 3 max-pooling take their place. Then, the first inception module comes forward. In this work, each inception layer is used with dimension reduction. Otherwise, convolutions would be computationally expensive. In each inception module, 1 × 1 convolution is first applied before 3 × 3 and 5 × 5 convolutions to reduce the computations. However, 3 × 3 max-pooling is first applied before 1 × 1 convolution, as
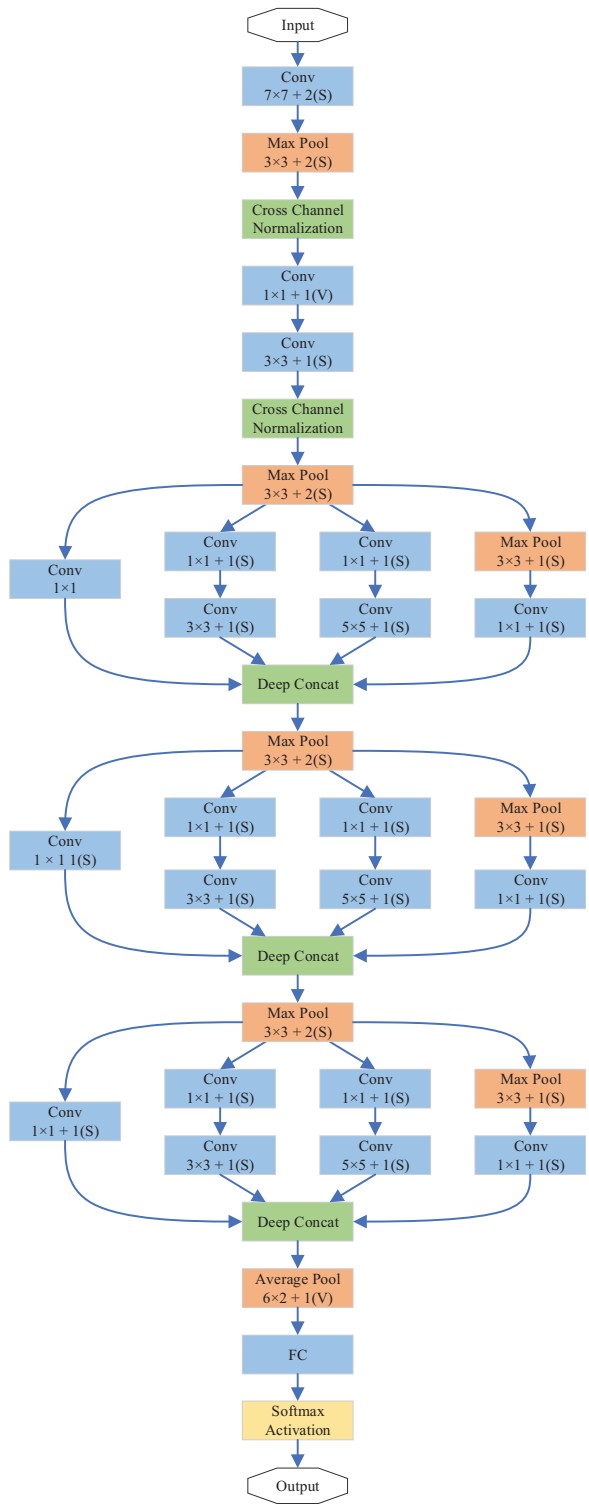

**Figure 4 Overview of the layers of the DiatomNet architecture.**

**Table 4  Comparison of DiatomNet with the other CNN models.**

| Model | Depth | Input image size |
|---|---|---|
| AlexNet | 8 | 227 × 227 |
| DiatomNet | 10 | 432 × 128 |
| GoogleNet | 22 | 224 × 224 |
| Inceptionv3 | 48 | 299 × 299 |
| ResNet18 | 18 | 224 × 224 |
| VGG16 | 16 | 224 × 224 |
| Xception | 71 | 299 × 299 |

illustrated in Fig. 4. Then, the filter concatenation part comes, and an inception module is finalized.

A 3 × 3 max-pooling is used after the first and second inception modules, a 6 × 2 average pooling is used after the third inception module. This design provides more accurate results than using max-pooling all the time. After the third inception model, a 40% dropout layer comes. Next, a fully connected layer takes its place. The network output is then calculated using the softmax activation function. The output size is 1 × 1 × 68 since we have 68 species of diatoms in our dataset. To train the network, rectified linear activation function (ReLU) is used for the fully-connected and convolutional layers. The depths and input image sizes of DiatomNet and the other CNNs are shown in Table 4 comparatively.

## EXPERIMENTAL WORK

In our work, the experiments were conducted in two stages. In the first stage, DiatomNet and the other CNN models were trained from the scratch and tested on both the original and augmented datasets as previously depicted in Table 2. The training and validation parts of the datasets were used to train the models, whereas the test parts were employed to evaluate the performances of the models on unseen data. The Glorot (also known as Xavier) (*Glorot & Bengio, 2010*) initializer was used to define the initial parameters of the models so that each weight was initialized with a small Gaussian value with zero mean and variance based on the fan-in and fan-out of the weight.

In the second stage, the transfer learning approach was used with the pre-trained versions of the existing CNN models. In each stage, the performances of the models were evaluated and compared in terms of several success metrics, including accuracy, precision, recall, F-measure, number of learnable parameters, training time, and prediction time. Due to the unbalanced structure of the dataset, weighted averages of accuracy, precision, recall, and F-measure were considered for comparisons. The experimental setup of the diatom classification system is illustrated in Fig. 5.

Input images were resized according to the input layer of each CNN model. Output layers of the CNN models were adapted to the number of classes (68) of the diatom image dataset. For all models, the mini-batch size was chosen as 16. Stochastic Gradient Descent (SGD) was used as the learning function with a learning rate of 0.0001 and a momentum of 0.9. In this learning function, the momentum helps accelerate the gradient vectors in the

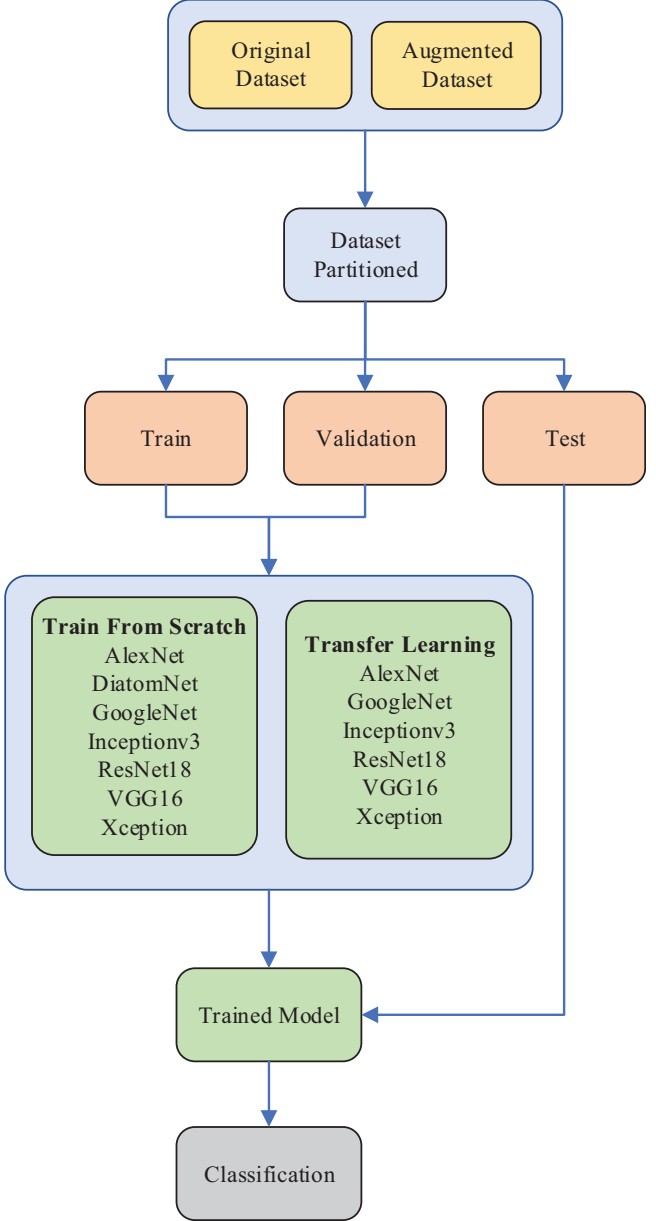

**Figure 5 The experimental setup of the diatom classification system.**

right directions to enable faster convergence. The models were then trained for 300 epochs from scratch, whereas 20 epochs were used for transfer learning. In all cases, early stopping was used to avoid overfitting. The experiments were conducted on a computer with a 3.79 GHz Intel Core i7 CPU, 32 GB of RAM, and a GeForce RTX 2060 Super GPU with 8 GB of RAM. All models were implemented using MATLAB 2020b Deep Learning Toolbox.

The results of each stage are elaborated on and discussed in the following subsections. Also, a comparison with the literature is provided.

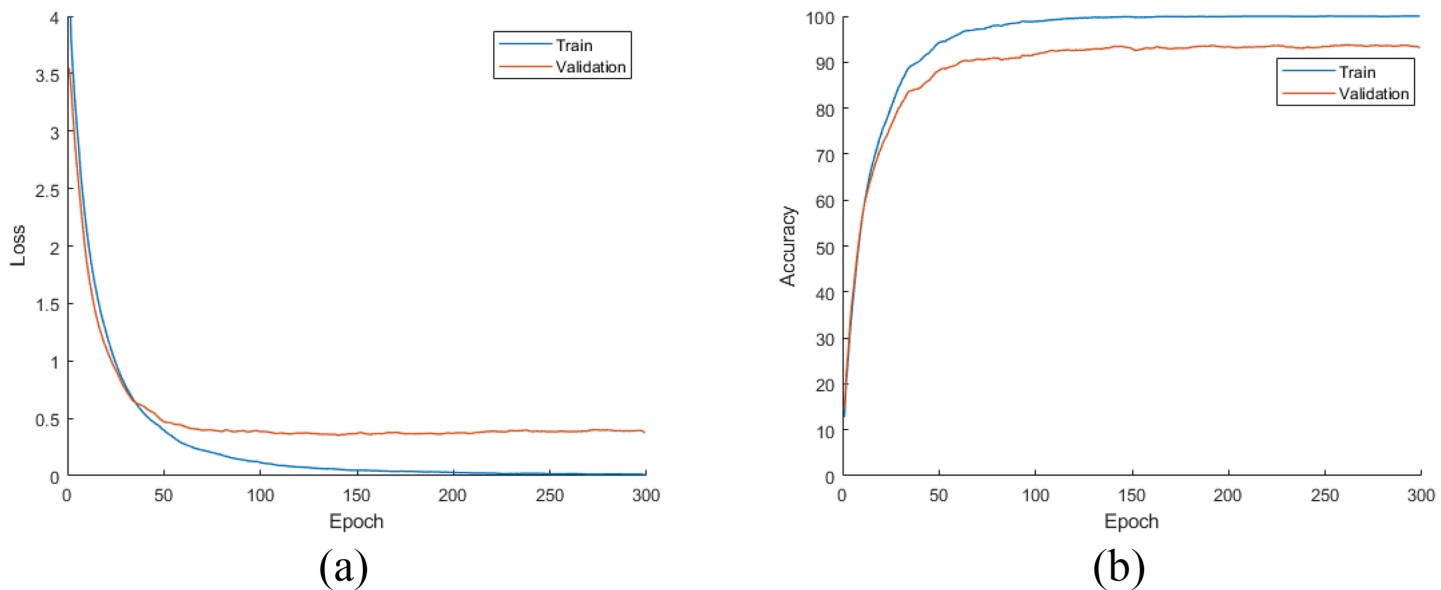

**Figure 6** (A) Loss and (B) accuracy plots for the training phases of the DiatomNet model on the original dataset.

## Results: DiatomNet *vs.* other CNNs

In this stage of the experimental work, DiatomNet and the other CNN models were trained from the scratch using the training and validation parts of the original dataset. The loss and accuracy plots for the training phase of the DiatomNet model are given in Figs. 6A and 6B, respectively. Then, the trained models were evaluated on the test part of the original dataset to find their performances on unseen data. As a result of this evaluation, the accuracy, recall, precision, and F-measure values of all models are comparatively shown in Fig. 7. As shown in this figure, DiatomNet, AlexNet, and GoogleNet achieved quite similar performances in terms of all metrics and surpassed the other models. In this experiment, the maximum accuracy, recall, precision, and F-measure values were all approximately 0.93. In contrast, the minimum values for the same metrics were achieved by the Xception model as 0.89, 0.89, 0.86, and 0.87, respectively.

Later, all CNN models were trained from scratch using the training and validation parts of the augmented dataset, which is four times largest than the original dataset as mentioned earlier. For this experiment, the loss and accuracy plots for the DiatomNet model's training phase are given in Figs. 8A and 8B, respectively. It is clear from these figures that all models were trained without overfitting. Then, the trained models were evaluated on the test part of the augmented dataset to reveal their performances on unseen data. As a result of this evaluation, the accuracy, recall, precision, and F-measure values of all models are comparatively shown in Fig. 9. Thanks to the augmentation, all models' performances were much better compared to the previous experiment with the original dataset. This time, DiatomNet offered the best performance in terms of all success metrics, while the performances of the other models were close to but slightly lower than

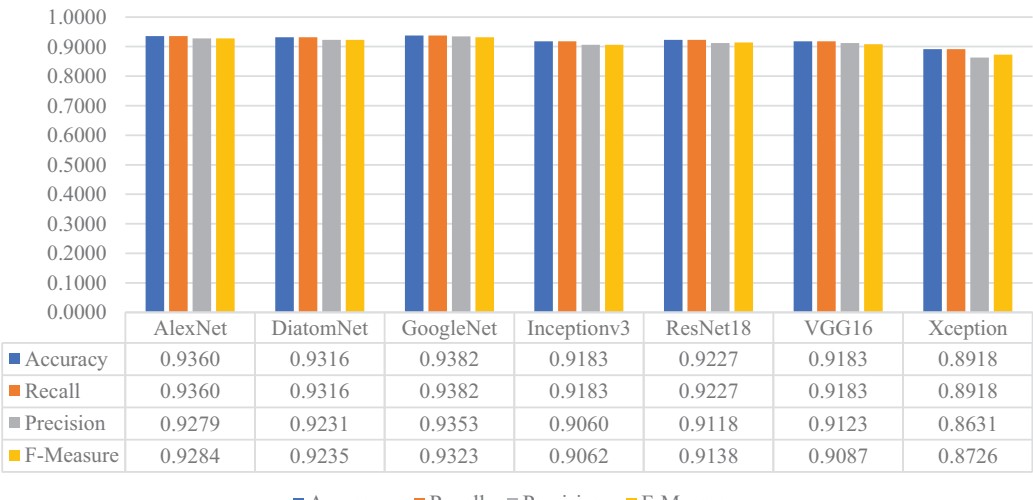

| | AlexNet | DiatomNet | GoogleNet | Inceptionv3 | ResNet18 | VGG16 | Xception |
|---|---|---|---|---|---|---|---|
| ■ Accuracy | 0.9360 | 0.9316 | 0.9382 | 0.9183 | 0.9227 | 0.9183 | 0.8918 |
| ■ Recall | 0.9360 | 0.9316 | 0.9382 | 0.9183 | 0.9227 | 0.9183 | 0.8918 |
| ■ Precision | 0.9279 | 0.9231 | 0.9353 | 0.9060 | 0.9118 | 0.9123 | 0.8631 |
| ■ F-Measure | 0.9284 | 0.9235 | 0.9323 | 0.9062 | 0.9138 | 0.9087 | 0.8726 |

■ Accuracy  ■ Recall  ■ Precision  ■ F-Measure

**Figure 7 Performances of the CNN models on the original test dataset.**

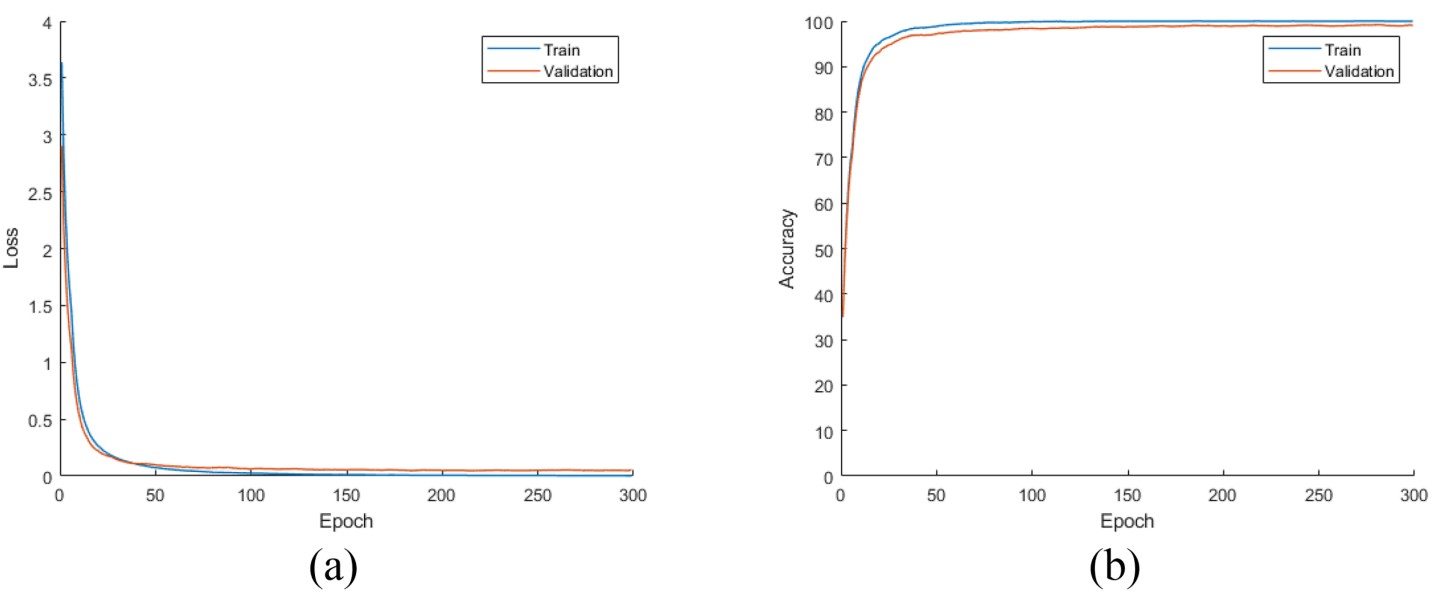

(a)               (b)

**Figure 8 (A) Loss and (B) accuracy plots for the training phases of the CNN models on the augmented dataset.**

DiatomNet. In this experiment, the maximum accuracy, recall, precision, and F-measure values were approximately 0.99, whereas the minimum values for the same metrics were around 0.97.

The performance of DiatomNet was also compared to the other models considering the number of learnable parameters, training time, and prediction time. The results of this comparison are summarized in Table 5. As shown in the table, the number of learnable

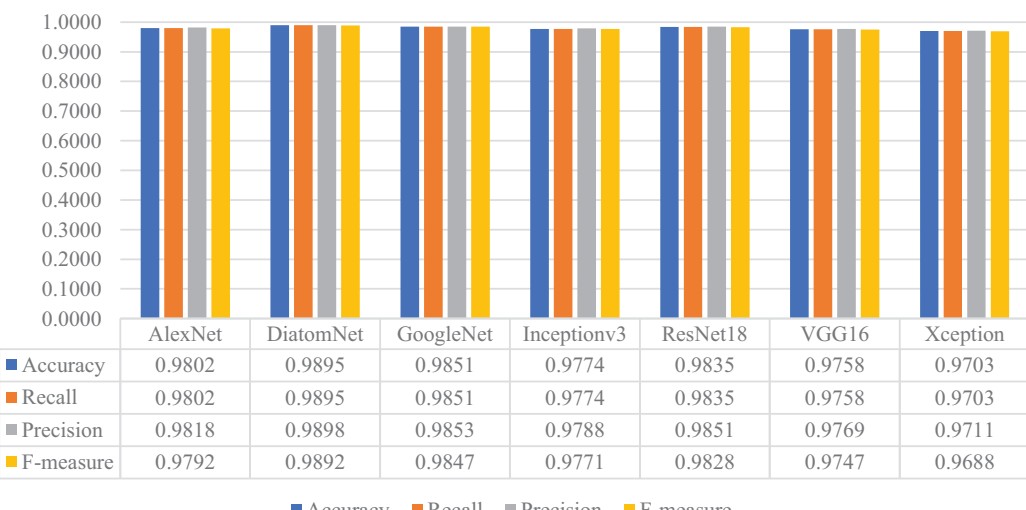

| | AlexNet | DiatomNet | GoogleNet | Inceptionv3 | ResNet18 | VGG16 | Xception |
|---|---|---|---|---|---|---|---|
| ■ Accuracy | 0.9802 | 0.9895 | 0.9851 | 0.9774 | 0.9835 | 0.9758 | 0.9703 |
| ■ Recall | 0.9802 | 0.9895 | 0.9851 | 0.9774 | 0.9835 | 0.9758 | 0.9703 |
| ■ Precision | 0.9818 | 0.9898 | 0.9853 | 0.9788 | 0.9851 | 0.9769 | 0.9711 |
| ■ F-measure | 0.9792 | 0.9892 | 0.9847 | 0.9771 | 0.9828 | 0.9747 | 0.9688 |

■ Accuracy   ■ Recall   ■ Precision   ■ F-measure

**Figure 9 Performances of the CNN models on the augmented test dataset.**

**Table 5 Complexities of the CNN models.**

| Model | Learnable parameters (millions) | Prediction time per image (ms) | | Training time per epoch (s) | |
|---|---|---|---|---|---|
| | | Original dataset | Augmented dataset | Original dataset | Augmented dataset |
| AlexNet | 44.56 | 0.3331 | 0.3385 | 19 | 58 |
| DiatomNet | 1.85 | 0.3213 | 0.3223 | 15 | 54 |
| GoogleNet | 6.04 | 0.5471 | 0.5507 | 27 | 96 |
| Inceptionv3 | 21.91 | 0.7072 | 0.6860 | 94 | 351 |
| ResNet18 | 11.21 | 0.4153 | 0.4190 | 34 | 128 |
| VGG16 | 113.57 | 2.2139 | 2.3018 | 160 | 584 |
| Xception | 20.94 | 1.3322 | 1.1116 | 101 | 382 |

parameters of DiatomNet (1.85 M) is far (up to 61 times) less than those of the other models. Thanks to this advantage and its architecture, DiatomNet achieved the shortest training time/epoch (54 s) and prediction time/image (0.32 ms) for the augmented dataset. The training time/epoch and prediction time/image of DiatomNet were up to 7 and 11 times shorter than the other models.

### Feature map analysis of DiatomNet

In this section, feature maps (class activation maps) were obtained to visualize which regions of diatom images contribute the most to the final classification decision (*Zeiler & Fergus, 2014*). This helps in understanding which parts of the images are crucial for the network to make a decision and provides insights into what features are indicative.

The feature maps associated with the early, intermediate, and deep layers of DiatomNet were derived using the sample images of each class presented in Fig. 2. While obtaining the feature maps, the channels exhibiting the maximum activations were considered. Through

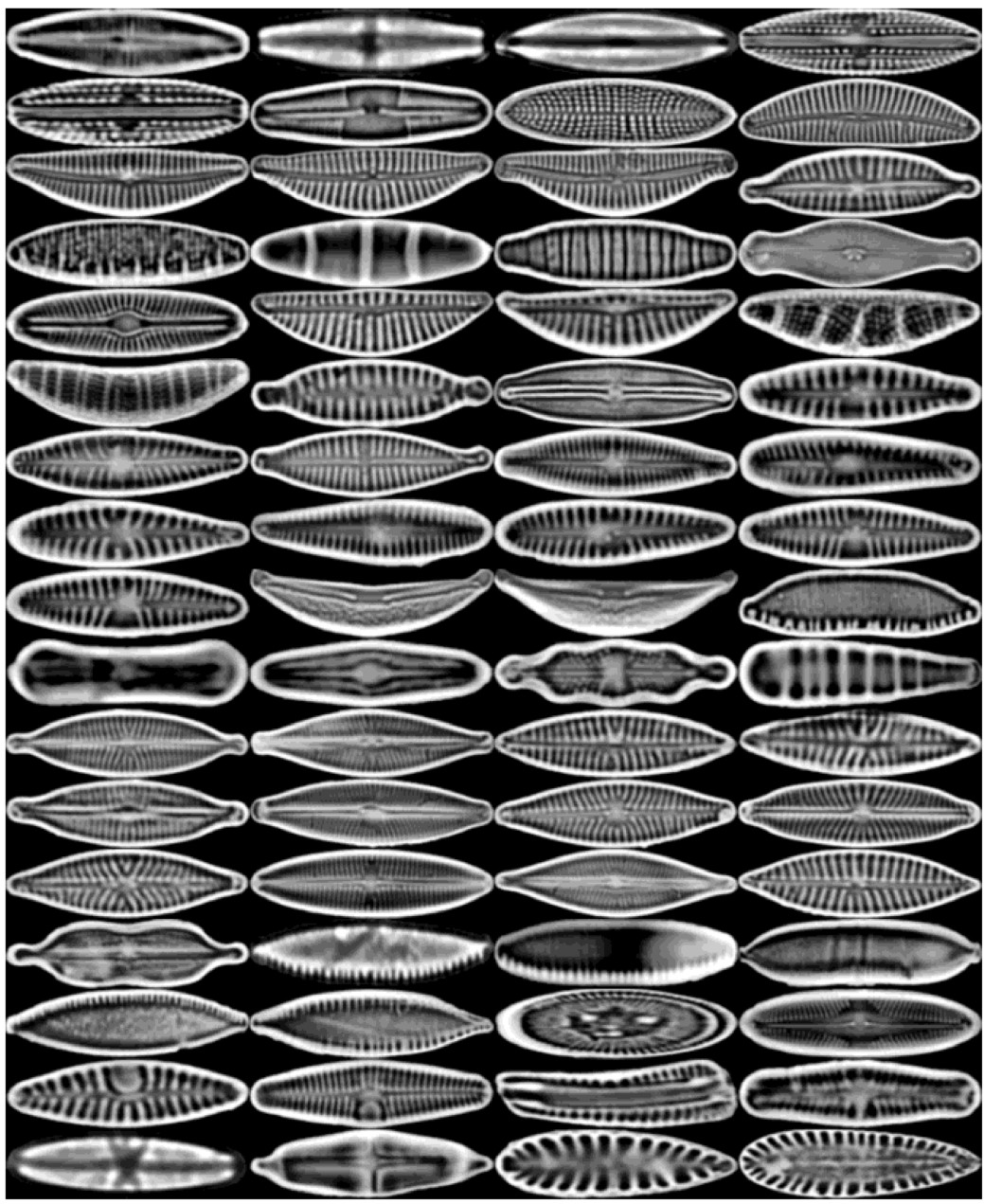

**Figure 10 The feature maps derived from the early layer of DiatomNet.**

a meticulous comparison of the activation regions with the corresponding regions in the original images, it was revealed which features the network had learned. Figure 10 shows the feature maps associated with the early layer of the DiatomNet network. These maps highlight basic and local features such as edges, color contrasts, and simple textures that are indicative of the presence of certain patterns in the diatom images.

The feature maps derived from the intermediate layer are shown in Fig. 11. The maps here capture mid-level features and patterns. Neurons in this layer respond to combinations of low-level features and represent parts of more intricate textures. Hence,

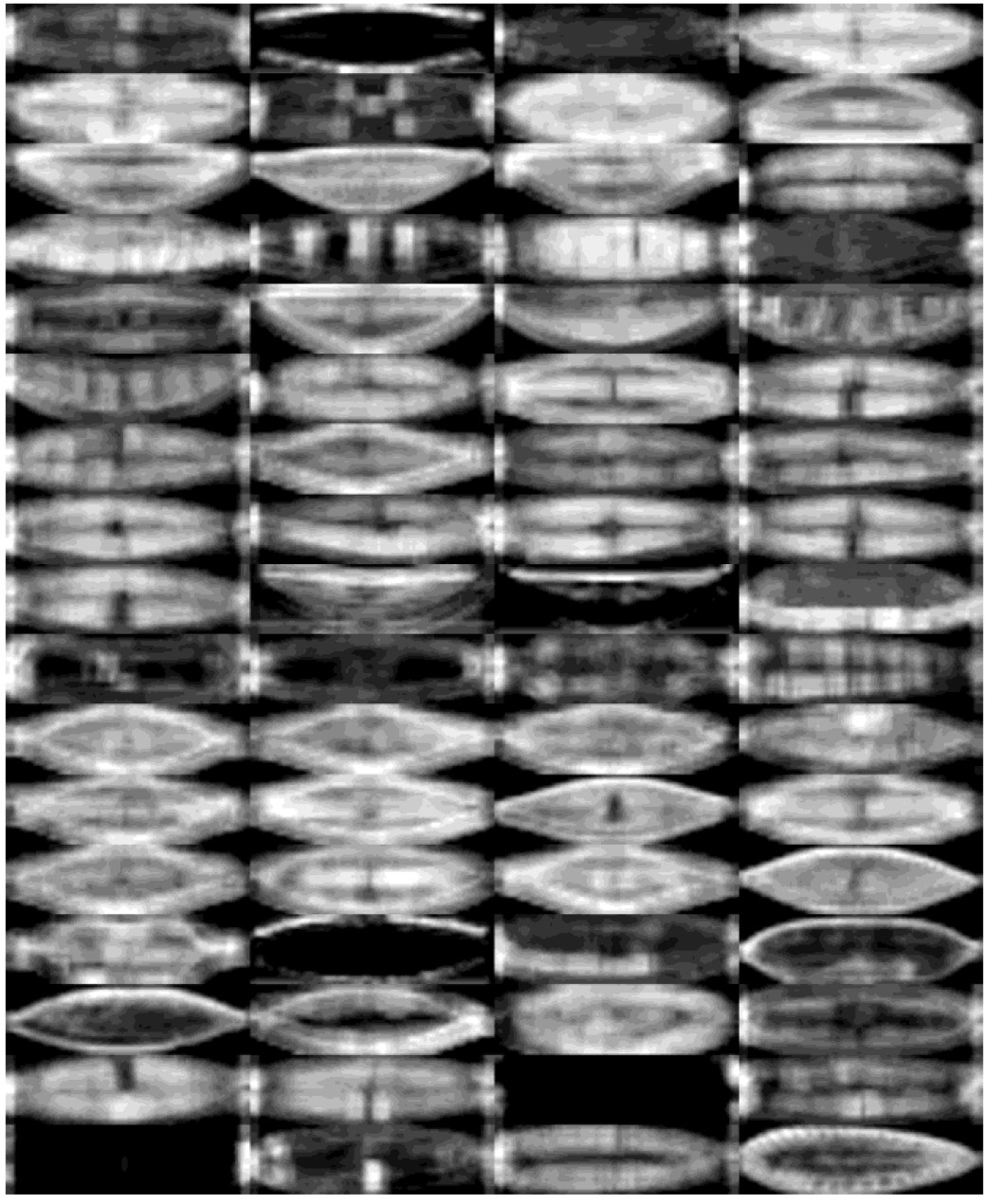

**Figure 11 The feature maps derived from the intermediate layer of DiatomNet.**

these maps help us identify specific structures within the diatom images that influence the classification of the network.

Figure 12 illustrates the feature maps associated with the deep layer of DiatomNet. These maps highlight high-level, abstract features of the diatom images that are crucial for the final classification. These features, hence, represent distinctive patterns associated with specific diatom classes.

Finally, heat maps corresponding to the feature maps of the deep layer are presented in Fig. 13. This visual representation serves a crucial role in enhancing our understanding of

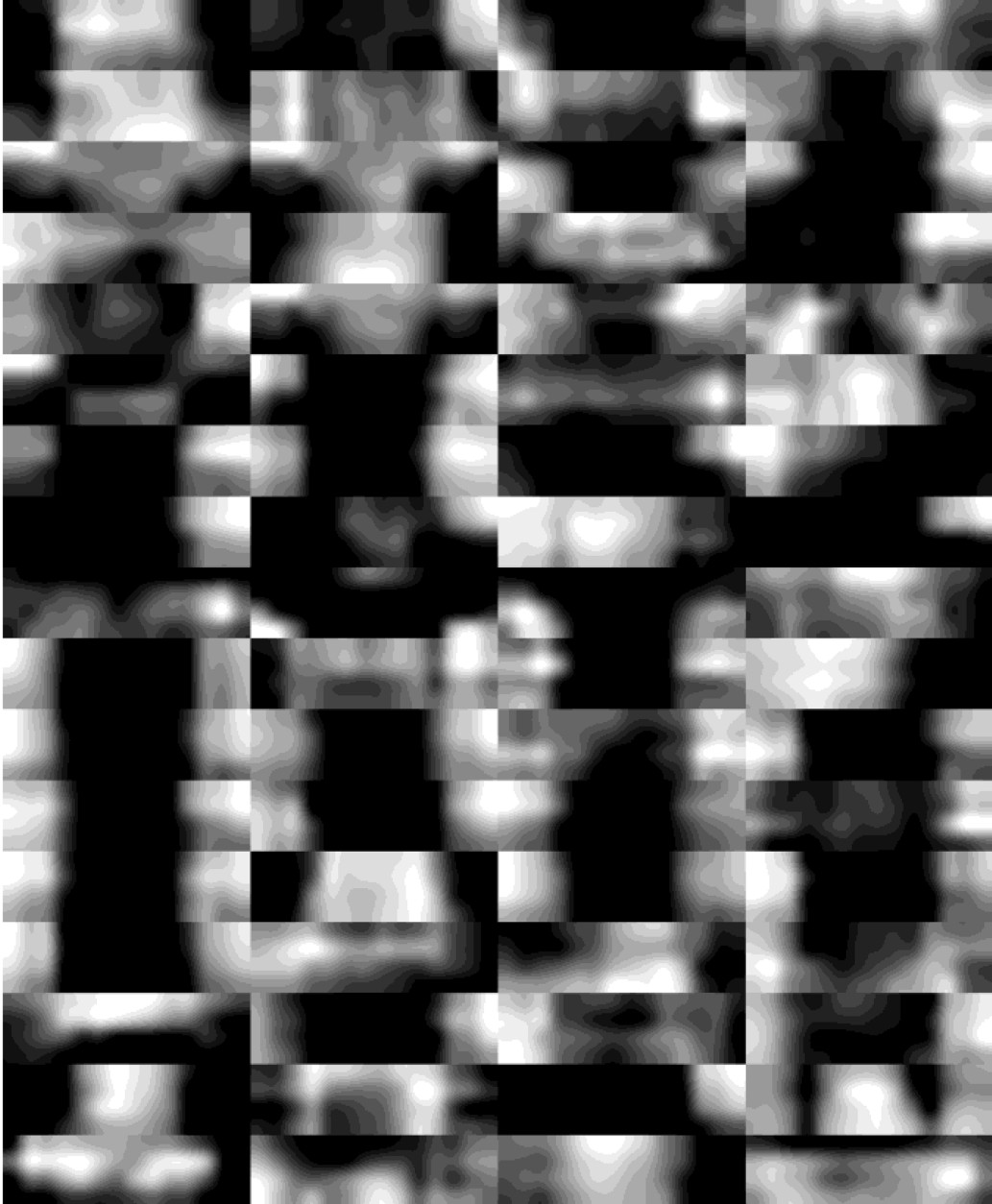

**Figure 12 The feature maps derived from the deep layer of DiatomNet.**

the pivotal regions within diatom images that play a significant role in classification. The inclusion of these heat maps enables the identification and localization of specific features and patterns within diatom images that exert the most influence on DiatomNet's output. Through the insights gained from these maps, one can readily discern the informative and relevant regions critical for discriminating between diatom classes, even in instances where the classes exhibit subtle similarities. The utilization of heat maps, therefore, proves instrumental in unraveling the discriminative power of the network, shedding light on the nuanced distinctions that contribute to its classification decisions.

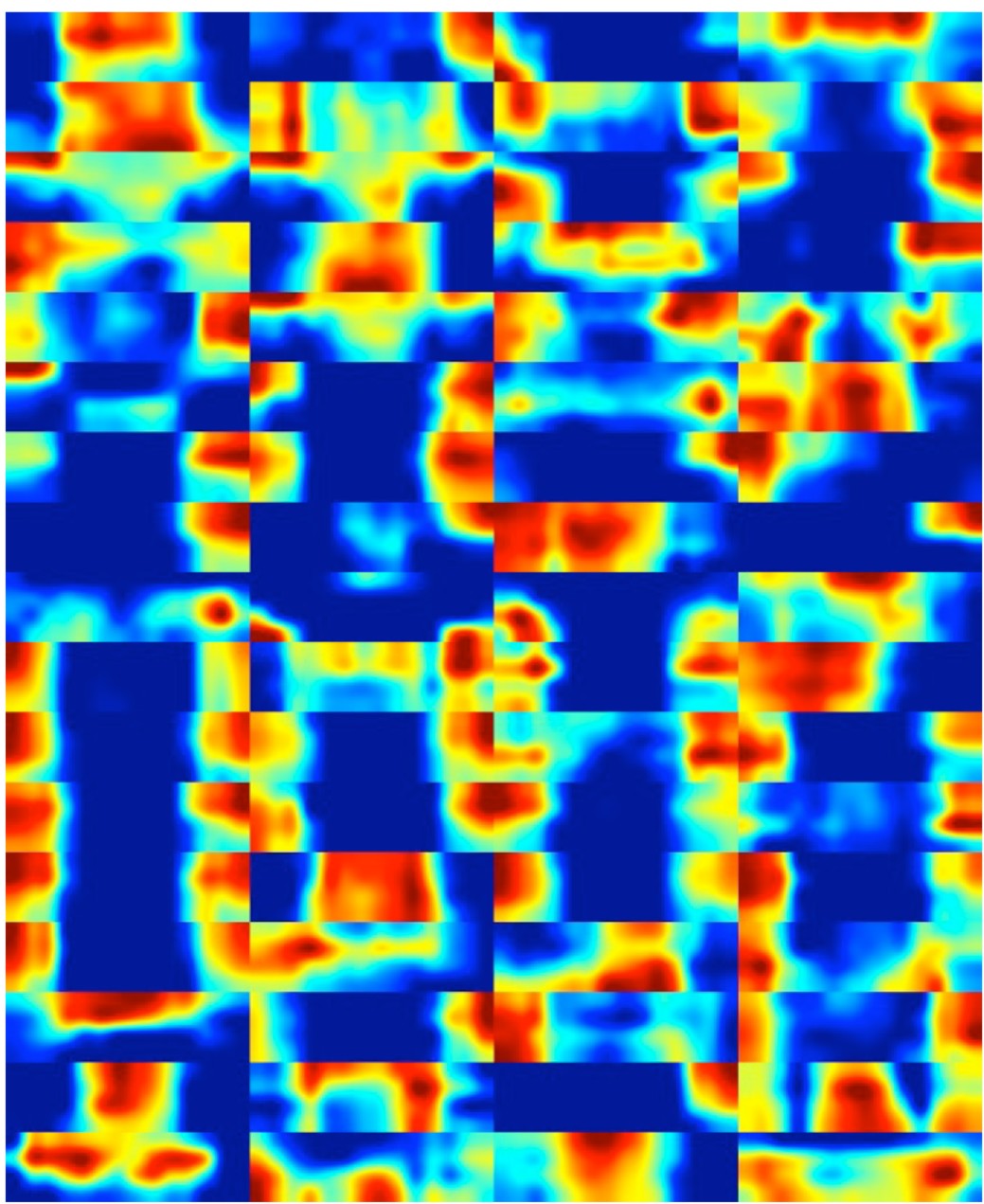

**Figure 13 The heat maps corresponding to the feature maps derived from the deep layer of DiatomNet.**

## Results: transfer learning

In the next stage of the experimental work, the pre-trained versions of the CNN models were re-trained and customized with the transfer learning approach using the training and validation parts of the original dataset. Then, the customized models were evaluated on the test part of the original dataset to assess their performances on unseen data. As a result, all models' accuracy, recall, precision, and F-measure values are shown in Fig. 14. As shown in this figure, VGG16 and Inceptionv3 achieved quite similar performances in terms of all

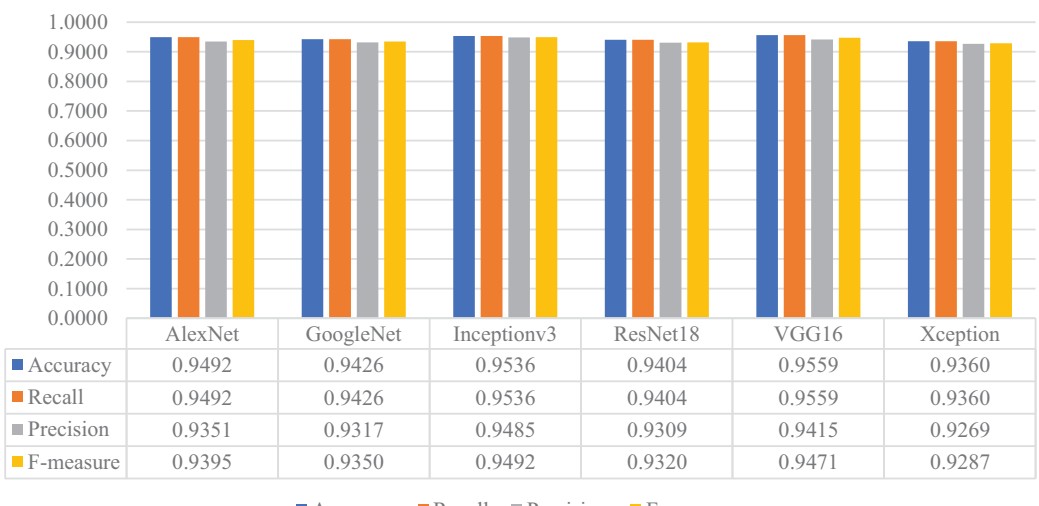

**Figure 14 Performances of the CNN models with transfer learning on the original test dataset.**

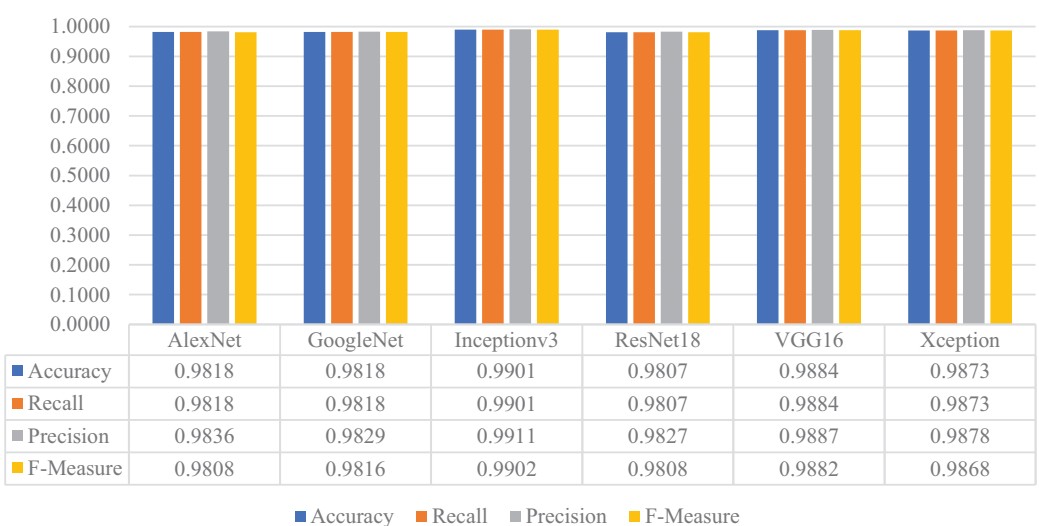

**Figure 15 Performances of the CNN models with transfer learning on the augmented test dataset.**

metrics and slightly surpassed the other models. For this experiment, the maximum accuracy, recall, precision, and F-measure values were around 0.95, whereas the minimum values for the same metrics were achieved with the Xception model at approximately 0.93.

Later, the pre-trained versions of the CNN models were re-trained with the transfer learning approach using the training and validation parts of the augmented dataset, instead of the original one. The customized models were then evaluated on the test part of the augmented dataset to assess their performances on unseen data. As a result, all models' accuracy, recall, precision, and F-measure values are shown in Fig. 15. Thanks to the augmentation, all models' performances were much better than the previous experiment with the original dataset. This time, Inceptionv3 got the best performance in all success

**Table 6 Complexities of the CNN models with transfer learning.**

| Model | Learnable parameters (millions) | Prediction time per image (ms) | | Training time per epoch (s) | |
|---|---|---|---|---|---|
| | | Original dataset | Augmented dataset | Original dataset | Augmented Dataset |
| AlexNet | 44.56 | 0.2894 | 0.2451 | 21 | 57 |
| GoogleNet | 6.04 | 0.4335 | 0.3796 | 27 | 90 |
| Inceptionv3 | 21.91 | 0.9965 | 0.9386 | 115 | 432 |
| ResNet18 | 11.21 | 0.3067 | 0.2417 | 22 | 80 |
| VGG16 | 113.57 | 1.6758 | 1.6971 | 109 | 541 |
| Xception | 20.94 | 2.2419 | 1.8831 | 142 | 815 |

metrics, whereas the other models' performances were similar and slightly lower than Inceptionv3. In this experiment, the maximum accuracy, recall, precision, and F-measure values were approximately 0.99, whereas the minimum values for the same metrics were around 0.98.

The performances of the models with transfer learning were also compared based on the number of learnable parameters, training time, and prediction time. The results of this comparison are summarized in Table 6. As shown in the table, GoogleNet has the smallest number of learnable parameters among all. AlexNet achieved the shortest training time/epoch (57 s) and ResNet18 achieved the shortest prediction time/image (0.24 ms) for the augmented dataset.

Considering the abovementioned results, DiatomNet, which is trained from scratch, provided a classification performance comparable to or even better than most of the customized models with transfer learning. While providing such a performance, DiatomNet significantly surpassed the other models regarding the number of learnable parameters, training time, and prediction time.

## Comparison with the literature

Comparison of the proposed work with the literature is summarized in Table 7. As can be seen from this table, the number of species in the studies varies between 6 and 80. The number of images used varies between 78 and 160,000. While some studies measure the classification performance of diatoms with only one morphology (*Luo et al., 2010*), others have classified diatoms with a wide variety of morphologies (*Bayer & Du Buf, 2002*; *Dimitrovski et al., 2012*; *Bueno et al., 2017*; *Pedraza et al., 2017*; *Sánchez, Cristóbal & Bueno, 2019*; *Libreros et al., 2019*; *Chaushevska et al., 2020*). Earlier studies were based on the classification of various feature extraction methods with different classifiers (*Bayer & Du Buf, 2002*; *Luo et al., 2010*; *Dimitrovski et al., 2012*; *Bueno et al., 2017*; *Sánchez, Cristóbal & Bueno, 2019*), state art of methods based on CNNs, which used well-known pre-trained networks (*Pedraza et al., 2017*; *Libreros et al., 2019*; *Chaushevska et al., 2020*). In some studies, only the original dataset was used (*Bayer & Du Buf, 2002*; *Luo et al., 2010*; *Dimitrovski et al., 2012*; *Sánchez, Cristóbal & Bueno, 2019*; *Libreros et al., 2019*; *Chaushevska et al., 2020*), while both the original and augmented datasets were used

**Table 7 Comparison of the proposed work with the literature.**

| Reference | # Species | # Images | Features | Classifier | Accuracy | Precision | Recall | F-measure |
|---|---|---|---|---|---|---|---|---|
| *Bayer & Du Buf (2002)* | 37 | 781 | Geometrical, textural, morphological, and frequency | Bagging Tree | 0.9690 | – | – | – |
| *Luo et al. (2010)* | 6 | 78 | Texture | BP neural network | 0.9600 | – | – | – |
| *Dimitrovski et al. (2012)* | 38 | 837 | Morphological, Texture | Random forest | 0.9797 | – | – | – |
| | 48 | 1,019 | | | 0.9715 | – | – | – |
| | 55 | 1,098 | | | 0.9617 | – | – | – |
| *Bueno et al. (2017)* | 80 | 24,000 | Morphological, statistical, textural, space-frequency | Bagging tree | 0.9810 | – | – | – |
| *Pedraza et al. (2017)* | 80 | 24,000 | AlexNet | Softmax | 0.9562 | – | – | – |
| | | 160,000 | | | 0.9951 | – | – | – |
| *Sánchez, Cristóbal & Bueno (2019)* | 8 | 703 | Elliptical fourier descriptors, phase congruency descriptors, gabor filter | Supervised: k-NN, SVM, Unsupervised: K-means, hierarchical agglomerative clustering, BIRCH | 0.9900 | – | – | – |
| *Libreros et al. (2019)* | – | 365 | GoogleNet | Softmax | 0.9200 | 0.8400 | 0.9800 | 0.9000 |
| | | | ResNet | | 0.8900 | 0.6000 | 0.6700 | 0.6300 |
| | | | AlexNet | | 0.9900 | 0.8400 | 0.9500 | 0.8900 |
| *Chaushevska et al. (2020)* | 55 | 1,100 | Inceptionv3 | Bagging random forest SVM Fine-tuned CNN | 0.8027 0.8636 0.9109 0.9872 | – | – | – |
| Proposed work | 68 | 12,108 | AlexNet | Softmax | 0.9802 | 0.9818 | 0.9802 | 0.9792 |
| | | | DiatomNet | | 0.9895 | 0.9898 | 0.9895 | 0.9892 |
| | | | GoogleNet | | 0.9851 | 0.9853 | 0.9851 | 0.9847 |
| | | | Inceptionv3 | | 0.9774 | 0.9788 | 0.9774 | 0.9771 |
| | | | ResNet18 | | 0.9835 | 0.9851 | 0.9835 | 0.9828 |
| | | | VGG16 | | 0.9758 | 0.9769 | 0.9758 | 0.9747 |
| | | | Xception | | 0.9703 | 0.9711 | 0.9703 | 0.9688 |
| | | | TL with AlexNet | | 0.9818 | 0.9836 | 0.9818 | 0.9808 |
| | | | TL with GoogleNet | | 0.9818 | 0.9829 | 0.9818 | 0.9816 |
| | | | TL with Inceptionv3 | | 0.9901 | 0.9911 | 0.9901 | 0.9902 |
| | | | TL with ResNet18 | | 0.9807 | 0.9827 | 0.9807 | 0.9808 |
| | | | TL with VGG16 | | 0.9884 | 0.9887 | 0.9884 | 0.9882 |
| | | | TL with Xception | | 0.9873 | 0.9878 | 0.9873 | 0.9868 |

together in others (*Bueno et al., 2017*; *Pedraza et al., 2017*). Most of the previous studies do not use a standard success metric, instead only a few metrics such as precision (*Bayer & Du Buf, 2002*; *Luo et al., 2010*; *Bueno et al., 2017*; *Pedraza et al., 2017*; *Sánchez, Cristóbal & Bueno, 2019*), whereas others use standard metrics, such as precision, recall, F-measure (*Libreros et al., 2019*; *Chaushevska et al., 2020*).

In our study, we use 12,108 images with various morphologies. The original data set was used directly as well as augmented. The effect of augmentation on classification tasks has been well investigated. The dataset augmented with the original dataset was classified and its effects were examined. Besides, it is the first time to compare the success of well-known CNNs, trained from scratch, with pre-trained versions. While existing studies use well-known models, in this study, a new model which is called DiatomNet is proposed and compared with well-known CNNs in various aspects, such as precision, recall, F-measure, training, and prediction times.

The most successful model so far is *Pedraza et al. (2017)* with AlexNet and an accuracy of 0.9951. However, this model has 160,000 diatom samples. They reported an accuracy of 0.9562 with 24,000 samples with AlexNet. They used AlexNet with 44.56 million learning parameters.

The DiatomNet proposed in our work offers an accuracy of 0.9895 with 12,108 samples and has only 1.85 million parameters.

In previous studies, the authors did not mention training and prediction times. But our study has shown that DiatomNet is also successful in terms of training and prediction times. For AlexNet trained from scratch using the original and augmented dataset, the training times for each epoch were 19 and 58 s, respectively. The prediction times of each image were 0.3331 and 0.3385 milliseconds. For DiatomNet, the training time for each epoch was 15 and 54 s, and the prediction time for each image was 0.3213 and 0.3223 milliseconds, respectively.

Our most successful model is the pre-trained Inceptionv3 model with an accuracy of 0.9901. If we can increase the number of samples in the dataset, we can achieve more successful results with these models.

## CONCLUSIONS

In our work, DiatomNet, a lightweight CNN model with 1.85 M learnable parameters and a depth of 10, was proposed to classify diatom species with significantly high accuracy while requiring low computing resources. The performance of DiatomNet was compared with those of popular CNN models and their customized versions obtained with transfer learning on the pre-trained versions of these models. A recently introduced diatom image dataset was utilized to train and evaluate all models. The experiments were conducted with both the original dataset and its augmented version. While comparing the performances, various success metrics, including accuracy, precision, recall, F-measure, number of learnable parameters, training time, and prediction time. The results of the experimental work verified that DiatomNet outperforms not only the other CNN models but also their customized versions (with just a few exceptions) in terms of all metrics. The augmentation of the original dataset further improved the performance of all models, as expected. Moreover, the utilization of feature maps and heat maps in our experiments enhanced the interpretability of the features derived from different layers of DiatomNet by highlighting the salient regions within diatom images that contribute most to the model's output. As a result, DiatomNet has proven to be not only a lightweight but also a strong candidate for

automated diatom classification tasks. In future work, DiatomNet would be tested on different datasets and its architecture would be further improved.

## ACKNOWLEDGEMENTS

We would like to thank Prof. Dr. Cuneyd Nadir SOLAK for his contributions to the diatom image dataset.

### Funding

The authors received no funding for this work.

### Competing Interests

The authors declare that they have no competing interests.

### Author Contributions

- Huseyin Gunduz conceived and designed the experiments, performed the experiments, analyzed the data, performed the computation work, prepared figures and/or tables, authored or reviewed drafts of the article, and approved the final draft.
- Serkan Gunal conceived and designed the experiments, analyzed the data, performed the computation work, prepared figures and/or tables, authored or reviewed drafts of the article, and approved the final draft.

### Data Availability

The Diatomnet dataset is available at Kaggle: Hüseyin Gündüz. (2022). Diatom Dataset [Data set]. Kaggle. https://doi.org/10.34740/KAGGLE/DS/1187591.

### Supplemental Information

Supplemental information for this article can be found online at http://dx.doi.org/10.7717/peerj-cs.1970#supplemental-information.

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
