# Peer review of "A lightweight convolutional neural network (CNN) model for diatom classification: DiatomNet"

_PeerJ Computer Science, doi:10.7717/peerj-cs.1970_

## Round 0.1 · original submission · Major Revisions

Based on the review reports, the paper needs a major revision.

Reviewer 1 ·

Basic reporting

The manuscript is well orginised.

Experimental design

In the 3.1 Result:DiatomNet vs. Other CNNs
The model was trained using the original dataset,which we could call it m1.Then another model was trained using augmented dataset which we could call it m2.
In the 3.2 Result:Transfer Learning
Did the pre-trained model mean m1 ?Later the re-trained model,which we could call it m3, was trained using the augmented dataset,and evaluated on the augmented dataset.
The performance of m2 and m3 was different.Could the ahthor give the reasons?

Why the m2 and m3 didn't evaluated on the original dataset like m1?

How did the author set the parameters of other CNN models ?

Validity of the findings

The conclusion is well stated.

·

Basic reporting

Good review. Code is included. The literature could also include other image networks.

Experimental design

It is unclear why other image segmentation models were not considered. Why CNNs? What about comparisons to graph network models? In general, it would be nicer to see how one can extract features and use it on new diatoms.For the annotations in Figure 1, it seems like these are incredibly simple shapes.

Validity of the findings

The models definitely show the classic overtraining signal, where training losses go to 0 (accuracy to 100%) with the validation saturating too.


Additionally, since the splits are per class, it is wholly unclear how this would perform on other (small sample) new data.

The model's baseline accuracy is suspiciously high as well. Given the simplicity of the shapes of the "diatoms" it seems like the model is learning fairly arbitrary representations.

There is no attempt made to discuss / view what the layers are learning.

Additional comments

I think with more insight this can find a place in the literature, given that this seems to be a popular system of study, however, most of the literature cited are conferences, which do not have the same quality standards. In particular, note that a novel data / architecture found experimentally without any theory (post-hoc or predesigned) is *not sufficient*. It is *necessary* to include a feature wise explanation.

---

## Round 0.2 · Minor Revisions

The writing should be improved. The motivation and applicability of the proposed research should be highlighted.

Reviewer 1 ·

Basic reporting

no comment

Experimental design

Experimental section is sufficient

Validity of the findings

The method can compare withthe latest methods.

Additional comments

There is an issue with the writing style of the paper, it feels like a splicing module. You can describe your job more.

·

Basic reporting

No comment.

Experimental design

No comment.

Validity of the findings

No comment.

Additional comments

Personally, I still find it strange to deploy larger models for diminishing returns. However, it shows a clear advantage over existing studies, and there is an attempt to justify the motifs learned, so it will suffice. It would be very nice to add a measure of "return on investment" in terms of model complexity increase (number of parameters) and the gain in predictor performance.

---

## Round 0.3 · accepted · Accept

The paper is ready to be accepted. Congratulations!

·

Basic reporting

No comment.

Experimental design

No comment.

Validity of the findings

No comment.

Additional comments

N/A.